# Eight-Week Pilates or Whole-Body High-Intensity Interval Training Program Improves Spinal Range of Motion During the Gait Cycle in Sedentary Women: A Preliminary Study

**DOI:** 10.3390/ijerph22020162

**Published:** 2025-01-26

**Authors:** Sabrina Fernandes Gonçalves, Arthur Ferreira do Vale, Cauê Vazquez La Scala Teixeira, Joyce Sousa de Oliveira, Jordana Rodrigues Vitória, Juliana Alves Carneiro, Mário Hebling Campos

**Affiliations:** 1Human Movement Assessment Laboratory, Faculty of Physical Education and Dance, Federal University of Goiás, Goiânia 74690-900, GO, Brazil; joycesousadeoliveira@gmail.com (J.S.d.O.); jordanavpersonal@gmail.com (J.R.V.); juliana.carneiro@ufg.br (J.A.C.); 2Faculty of Physical Education and Dance, Federal University of Goiás, Goiânia 74690-900, GO, Brazil; arthur_vale27@hotmail.com; 3Funcional Link, Santos 11070-001, SP, Brazil; caue_jg@yahoo.com.br

**Keywords:** spinal curvature, Pilates, WBHIIT, sedentary behaviour

## Abstract

This study aimed to compare the effects of Pilates (PIL) and whole-body high-intensity interval training (WBHIIT) on the spinal curvature of sedentary women. After being invited, 26 sedentary women aged between 20 and 54 voluntarily agreed to participate in the study. The sample was obtained through convenience sampling, and the participants chose either PIL or WBHIIT, which resulted in 13 participants in each group. Spinal posture was assessed pre- and post-intervention through videogrammetry during standing and walking. Markers were placed on the back, and the volunteers were instructed to remain in a standing position on a stationary treadmill for ten seconds. Subsequently, the treadmill was activated at a speed of 5 km/h. After one minute of walking, a complete gait cycle was recorded for analysis. The results showed no significant changes in spinal angles in static position between groups. However, in the walking position, there was a large-magnitude increase in the spinal range of motion (ROM) post-intervention (PIL Lumbar *d* = 1.8; PIL Thoracic *d* = 2.9; WBHIIT Lumbar *d* = 1.0; WBHIIT Thoracic *d* = 3.5) within groups in the sagittal plane. The adaptations promoted by these interventions in spinal ROM are important for reducing the risks of spinal stiffness and pain due to sedentary behaviour.

## 1. Introduction

Sedentary behaviour is closely linked to various health issues, including cardiovascular diseases, diabetes, obesity, and notably, musculoskeletal disorders [1,2]. Prolonged physical inactivity exacerbates the risk of developing musculoskeletal problems, particularly lower back pain, which has a high prevalence among sedentary individuals, especially women [3,4,5]. This increase in lower back pain is often attributed to compromised posture and reduced range of motion (ROM) due to inactivity [5].

Women who lead sedentary lifestyles are particularly vulnerable to postural problems, with a higher incidence of lower back pain reported in this population [5]. The reduction in ROM, especially in the spinal region, is one of the contributing factors to the onset of lower back pain [5,6,7] and altered patterns during gait [5]. These issues underscore the necessity for effective interventions to improve posture and spinal mobility, reducing the burden of lower back pain and musculoskeletal disorders in sedentary populations.

Physical exercises have been extensively studied for their potential to enhance spinal ROM and alleviate musculoskeletal discomfort [8,9,10]. Among these, Pilates has garnered considerable attention for its efficacy in promoting spinal strength, flexibility, and overall body awareness [11,12,13]. Pilates is well documented for its role in enhancing spinal mobility and strengthening core muscles, which are crucial for maintaining proper posture and preventing spinal pain [13].

In addition to Pilates, whole-body high-intensity interval training (WBHIIT), which involves bodyweight exercises performed at high intensity, has shown potential in improving various health-related factors, including cardiovascular health, body composition, and respiratory capacity [14,15,16]. Although WBHIIT is known for its general health benefits, its specific effects on postural adaptations and spinal health remain under-explored. The exercises typically included in WBHIIT programs (for example squats, mountains climbers, and sit ups) can induce postural adaptations, such as enhancing range of motion (ROM) and strengthening the core and back muscles [16].

Research suggests that both Pilates and WBHIIT could be beneficial in mitigating the adverse effects of sedentary behaviour on spinal posture [17,18,19,20,21]. However, while Pilates has a well-established evidence base supporting its role in enhancing core strength and spinal mobility, studies examining the postural adaptations resulting from WBHIIT are limited. WBHIIT is a cost-effective exercise modality that requires no specialised equipment or dedicated facilities, making it suitable for practice in public spaces such as parks and squares [22,23]. Understanding the comparative effects of these two exercise modalities on spinal curvature and ROM can help extend roll-off interventions aiming at reducing sedentary-related spinal issues in all age groups.

This study aims to compare the effects of Pilates and WBHIIT on spinal curvature and ROM in the sagittal plane of sedentary women. The objective is to determine the extent to which these exercise modalities can induce postural adaptations and improve vertebral mobility, thereby offering potential strategies for the prevention and management of muscular skeletal disorders related to posture in sedentary populations. It is hypothesised that both Pilates and WBHIIT will promote significant improvements in posture and spinal mobility, with observable benefits during static postures and dynamic activities such as gait.

## 2. Materials and Methods

### 2.1. Experimental Approach

This study is a non-randomised comparative clinical trial in which two eight-week interventions, whole-body high-intensity interval training (WBHIIT) and Pilates, were applied to sedentary woman. The sagittal vertebral posture in walking and standing positions was measured both pre- and post-exercise programs.

### 2.2. Participants

After being invited, 26 sedentary women aged between 20 and 54 voluntarily agreed to participate in the study. The participants chose either Pilates or whole-body high-intensity interval training (WBHIIT), which resulted in 13 participants in each group. Participants had no prior experience with the selected modalities.

The inclusion criteria were ages between 18 and 59, absence of regular exercise, and fitness for regular physical activity. Exclusion criteria included recent myocardial infarction (less than six months); recent or incapacitating stroke (less than six months); congestive heart failure; chronic renal failure; type 1 and/or uncontrolled diabetes; untreated or severe neuromuscular, musculoskeletal, or joint diseases, and any other disease or limitation that could compromise the execution of exercise protocols. All data collection was conducted in a laboratory. An independent sample t-test (*p* < 0.05) showed baseline similarity between groups (Table 1).

The activities were offered by the Center for Body Practices at the University (Table 1). All participants signed an informed consent form (ICF) before completing a form with demographic, social, and health information. The research project was approved and registered with the Research Ethics Committee (nº 3.337.411). This research is part of a larger study entitled “Acute and Chronic Effects of Different Types of Physical Exercise on Biodynamic and Subjective Health Parameters”. This project foresees other phases with a larger sample size and random distribution of participants into groups, for a powerful statistical design. The present work is the initial phase that used a convenient sample size. The interpretation of the results should consider these characteristics. Despite these limitations, the independent samples *t*-test (*p* < 0.05) showed baseline similarity between the groups (Table 1).

### 2.3. Interventions

#### 2.3.1. The Pilates Intervention Method

The Pilates intervention method was conducted over eight weeks, with 50 min sessions twice per week, totalling 16 sessions. The training sessions were group-based and conducted by students and professors from a Physical Education undergraduate course. The exercises followed the systematisation of Authentic Pilates from The Pilates Studio Brazil [15,17,20]. A Swiss ball was used as the modified Pilates equipment in this intervention to promote resistance or facilitate movement. Some trunk stabilisation exercises from modified Pilates were included in the sessions, primarily plank exercises with the trunk in different positions (dorsal, ventral, and lateral) with varying levels of execution difficulty, with and without the Swiss ball.

The classes were structured by time as follows: 5 min for body adaptation and familiarity with the floor; 20 min for teaching and learning Pilates techniques, focusing on breathing and powerhouse activation (classical Pilates exercises used the following: the hundred, roll over, one leg circle, rolling like a ball, single leg stretch, double leg stretch, single straight leg stretch, double straight leg stretch, criss-cross, double leg kicks, adapted shoulder bridge, and sidekicks); 15 min for trunk stabilisation exercises using the Swiss ball, predominantly plank exercises with the body in dorsal, ventral, and lateral positions; and finally, 10 min for stretching and relaxation exercises.

In the first three weeks, Pilates exercises were performed in one set of 3 to 10 repetitions for dynamic exercises and in isometric postures for 3 to 10 s, according to the individual’s perception of effort and comfort. From the fourth week onwards, a second set was added for all exercises. Additionally, the method’s intensity and progression increased by raising the level of difficulty, resistance, and range of motion of each exercise when optimal execution and control were achieved.

The exercises followed the application progression established by the classical Pilates method, moving from basic to intermediate exercises and adapting to everyone’s physical limits, execution ability, and range of motion. Exercise execution adhered to the method’s principles: breathing, control, concentration, centring, precision, flow, coordination, and postural alignment [15,17].

#### 2.3.2. The WBHIIT Protocol

The WBHIIT protocol was conducted over eight weeks and prescribed based on the study by Evangelista et al. [16]. During the first two weeks, 15 rounds of 30 s of high-intensity exercise (>9 on the Subjective Perception of Effort scale—SPE—or “all out”, as many repetitions as possible) were performed, followed by 30 s of passive recovery. From the third week, the protocol was modified to 20 rounds of 30 s of high-intensity exercise (>9 on SPE or “all out”), followed by 30 s of active recovery (SPE < 4) with a low-complexity, low-metabolic-demand exercise, such as walking or light jogging. The adapted Borg Scale for Subjective Perception of Effort, used in Brazil, was employed [24].

The exercises used during the WBHIIT stimulus phase predominantly included the following: jumping jacks, high-knee running in place, step climbing, squats, squat jumps, split squat, squat thrust, push-ups, burpees, mountain climbers, and jump alternating lunges. Although there are no established guidelines for exercise selection criteria in WBHIIT, according to Machado et al. [25], intensity progression can be managed through the complexity of motor task execution, that is, from simpler to more complex movements, and from single-pattern exercises to combined exercises. Additionally, progression involves moving from exercises with lower to higher cardiovascular, musculoskeletal, and energy expenditure demands. These strategies were viable for organising the protocols over the eight weeks. In WBHIIT, during the first two weeks, the 20 rounds of 30 s exercises were performed with less complex and lower energy expenditure exercises such as air squats, going up and down a step, jumping jacks, high-knee running in place, and sit-ups. In the following three weeks, there was an intensity progression, and the 30 s exercises included squat thrusts, push-ups, mountain climbers, and walking lunges. Finally, in the last three weeks, the exercises progressed to include squat jumps, squat thrusts, push-ups, mountain climbers, jump alternating lunges, and burpees.

The training session lasted 40 min, including a 10 min warm-up with low-impact aerobic exercises of moderate intensity, such as brisk walking or running in place, and a 10 min cool-down with static stretching for the entire body. A total of 16 WBHIIT sessions were completed over eight weeks.

### 2.4. Postural Assessment

The participants wore swimming caps, a top with narrow straps on the back and running or legging shorts. Retro-reflective markers (12 × 8 mm) were adhered to the skin on the back [26]. Markers were placed at the spinous process of the second sacral vertebra (S2), fourth lumbar vertebra (L4), and the 1st (T1), 6th (T6), and 12th (T12) thoracic vertebrae. Two pairs of markers aligned with both posterior superior iliac spine were used as references for analysis. One pair was fixed at the level of the spinous processes of the T6, and another pair at L4. After marking, the volunteers were instructed to remain in a standing position on a stationary treadmill (Movement LX150) for ten seconds. Subsequently, the treadmill was activated at a speed of 5 km/h, and the participants began walking. After one minute of walking, a complete gait cycle was recorded for analysis.

Images of the back and control points with a known location (calibration) were captured using three OptiTrack Flex 13 cameras (Natural Point Inc., Corvallis, OR, USA), synchronised via hardware and operated by Motive software (version 2.3.1), with the following image acquisition parameters: frequency of 100Hz, Exposure 500, THR 200, and 15 LEDs. Dynamic Posture [27,28], a software developed in the MatLab^®^ (The MathWorks, Natick, MA, USA), was used for image processing to track the barycentre [26] of the retro-reflective markers, and its three-dimensional (3D) reconstruction using the direct linear transformation (DLT) method [29]. The global reference frame of the laboratory was defined as follows: vertical axis Z (upward), posterior–anterior horizontal axis X (forward), and lateral horizontal axis Y (to the left).

The 3D coordinates of all markers were described in a local frame of reference on the trunk, originating at T12 [30]. The vector from L4 to T6 defined the orientation of the longitudinal axis z (upward). An auxiliary vector y’ was defined with its origin at the midpoint of the reference points to the right of L4 and T6 and the end at the midpoint of the reference points to the left of L4 and T6. The cross-product between y’ and z defined the orientation of the sagittal axis x (forward), and the cross-product between z and x defined the orientation of the transverse axis y (to the left). The positions of the spinal markers were projected onto the sagittal plane of the trunk (normal to y). The spine was modelled as four rigid segments for angular data computation (Figure 1). The lumbar sagittal angle (α) was quantified by the angle between the following two segments [27,28,30], projected onto the trunk local sagittal plane: the thoracolumbar segment was a straight line from L4 to T12; and the lumbosacral segment was from S2 to L4. The thoracic sagittal angle (β) was quantified by the angle between two segments projected onto the trunk sagittal plane: the superior thoracic segment was the straight line from T6 to T1; and the inferior thoracic segment was from T12 to T6. Positive values indicated anterior concavities (kyphosis), negative values indicated posterior concavities (lordosis), and null values represented a rectified spine.

One instant (frame) for the standing position and one complete gait cycle (two steps) in the last walking minute were tracked and analysed. The gait cycle was defined between two successive right foot strikes on the treadmill and was normalised in time to 101 points, representing positions from 0 to 100% of the standard gait cycle. For walking, the range of motion was calculated as the absolute difference between the highest and lowest values shown in the standard gait cycle. The neutral curve (average posture during gait) was also obtained [27].

The initial data collection, establishing the participants’ baseline/pre-intervention data, took place in April 2019. Following the training period (8 weeks) with the respective physical exercise modalities, the second round of data collection was conducted at the beginning of July 2019.

### 2.5. Statistics

The data showed a normal distribution (Shapiro–Wilk) and similar variances (Bartlett). Data were reported through means and standard deviation. An independent sample t-test was used for baseline (pre) analysis. A two-way ANOVA (groups × time) was used to compare the effect of the intervention. When a significant difference was detected, the Tukey–Kramer multiple comparison test was used for post hoc testing. In these cases, the effect size (Cohen’s *d*) was also calculated to determine the magnitude of differences between conditions (*d* < 0.2 = trivial, 0.2 ≤ *d* < 0.5 = small, 0.5 ≤ *d* < 0.8 = medium, *d* ≥ 0.8 = large effects). For effect size calculation, the difference between average values, pair by pair, was divided by the combined standard deviation of pre-intervention measures. Statistical significance was set at 5%, and version R2013b of Matlab^®^ (The MathWorks, Natick, Massachusetts, USA) was used.

The sample size calculation was performed post hoc using GPower software (version 3.1.9.6), employing an effect size (Cohen’s *d*) of 0.8, an alpha level (*α*) of 0.05, and a beta level (*β*) of 0.8.

## 3. Results

The post hoc sample size calculation resulted in a total sample of 20 participants. Based on the results obtained, the sample in this study (*n* = 26) appears to be of reasonable quality.

The results (Table 2) for the standing posture indicated that WBHIIT group had a more flexed (kyphosis) thoracic spine than Pilates group, before and after the intervention, and that situation was not affected by the interventions. However, lumbar angles in static position were similar between groups. Both lumbar and thoracic angles in static position showed no significant changes post-intervention.

Regarding gait (Table 2), there was no difference between groups for average angles as for range of motion. No differences were presented for average walking angles after interventions. However, both exercise modalities promoted a large magnitude increase in the range of motion of the lumbar and thoracic regions in the sagittal plane.

As shown in Figure 2 and Figure 3, there was a significant increase in the range of motion of the spine in the sagittal plane, both in the lumbar and thoracic regions, respectively, for both groups after the intervention.

## 4. Discussion

The present study aimed to identify adaptations in spinal posture, particularly potential changes in angles and range of motion in the sagittal plane, in healthy women who underwent 8 weeks of bodyweight training, comparing whole-body HIIT (WBHIIT) and Pilates interventions.

The results showed that both groups exhibited significant large-magnitude intra-group changes in range of motion (ROM) during gait after intervention. Increases in flexibility can be beneficial for individuals with limited mobility due to sedentary behaviour or postural disorders, such as lumbar rectification, which affect gait quality [31]. Galbusera and Bassani [32] highlight that spinal flexibility directly influences muscle activation patterns during gait, and muscle stiffness can increase the effort required to perform proper movements. Studies such as that of Mekhael et al. [33] indicate that spinal flexibility plays a crucial role in shock absorption during gait. A reduction in ROM, or its excessive enlargement, can compromise spinal biomechanics and increase the risk of injury [34,35]. Thus, for individuals with vertebral instability, exercises that excessively increase ROM may be harmful and require careful evaluation before practice.

A sedentary lifestyle significantly contributes to spinal stiffness and discomfort [36,37]. Previous studies, such as those by Kett and Sichting [36] and Schneider et al. [38], have demonstrated that long periods of inactivity can lead to postural deterioration and increased muscle stiffness, resulting in reduced range of motion (ROM) of the spine. Spinal ROM, according to Hamill [39] and Hall [40], is crucial for performing daily activities such as trunk flexion, extension, and rotation.

Muscle stiffness or inelasticity requires increased effort to perform movements, leading to less optimised gait patterns, especially when there is a reduction in the ROM. The increase in ROM observed in the participants of the Pilates group in this study would therefore contribute to improved gait stability and functional capacity. This effect was also observed in the study by Carneiro et al. [20], which demonstrated that a twelve-week intervention with the Pilates method in a group of sedentary obese women was able to increase the distance covered in the 6 min walk test, as well as flexibility in the sit-and-reach test, indicating an improvement in functional capacity and dynamic gait stability.

Regarding the intervention with WBHIIT, the results of our study suggest that incorporating high-intensity movements using bodyweight can be effective in improving spinal mobility and countering the effects of sedentary behaviour. As a time-efficient and accessible practice, WBHIIT can be an important strategy for promoting health and reducing sedentary behaviour, in line with WHO guidelines [41]. Exercises used in WBHIIT, such as squats and burpees, are known to improve joint mobility and promote health benefits, such as the prevention of cardiovascular diseases and increased respiratory capacity [21,22,23,25]. These features make WBHIIT a practical option to combat sedentary behaviour, especially as it does not require complex equipment [25].

Sedentary behaviour is strongly associated with muscle weakness, which can compromise spinal support [6,7], contributing to hyperkyphosis. This condition can result in a weakening of the spinal extensors and shortening of the pectoral muscles, leading to muscle imbalances and discomfort [3,5]. Another consequence of sedentary behaviour is muscle shortening, which can lead to altered gait patterns, reducing the ROM of the spine and worsening pain conditions [42].

Based on the findings of Will et al. [43] and Seyedhoseinpoor et al. [44], chronic lower back pain is associated with the contraction of the paravertebral muscles and adjacent structures. This muscular contraction acts similarly to an abdominal orthosis, restricting spinal flexibility and range of motion, which exacerbates the pain. Furthermore, individuals with chronic lower back pain often adopt compensatory postures that further limit spinal range of motion, as documented by Seyedhoseinpoor et al. [44].

The increase in ROM of the thoracic and lumbar regions in both groups demonstrates the effectiveness of the exercises proposed in this study in improving spinal flexibility and mobility. This underscores the importance of practices that include both mobility and strengthening, essential for maintaining quality of life and combating sedentary behaviour.

### Study Limitations

This is a preliminary study in which the sample was obtained through convenience sampling, the allocation of participants to groups was not randomised, and the sample size calculation was conducted post hoc. Although the groups were similar at baseline and the sample size appears to exceed the minimum required for a large effect size, the results should be interpreted with caution regarding generalisations.

This study is characterized as a non-randomised clinical trial, as the sampling was conducted for convenience, selecting participants enrolled in the Body Practices Center of FEFD/UFG. Clinical trials are widely recognised as the ideal approach for evaluating the effects of specific interventions, being used to generate data on the efficacy, safety, and effectiveness of treatments. However, randomisation, considered the gold standard in clinical trials, was not possible in this study.

As Minneci and Deans [45] (p. 335) argue, “when randomized trials are not feasible due to strong preferences that lead only a small proportion of patients to accept randomization, a high-quality non-randomized parallel group study, in which patients choose their treatment, is a valuable alternative”. These authors also advocate for clinical trials with smaller groups, where statistical randomisation is not feasible, as they can be equally valid, better representing minorities, ensuring higher participant adherence to the intervention, and enabling the evaluation of well-established treatments in the literature, replicating conditions closer to clinical practice.

In this study, videogrammetry was used to analyse spinal posture through markers attached to the skin. This methodology is widely used in biomechanical analyses and has evidence of validity [26,27,28,29,30,46]. Although videogrammetry does not provide direct measurement of the bone configuration and spinal structures, it is advantageous in reducing participant exposure to radiation, as seen with X-rays.

Additionally, no prior functional assessment or detailed diagnosis of participants’ postural deviations was conducted. In this sense, it is recommended to apply the methods of this study to specific populations with postural deviations, such as hyperlordosis, hyperkyphosis, or scoliosis, to verify the potential benefits of the proposed interventions.

## 5. Conclusions

Based on the observed results, it is concluded that both modalities, Pilates and whole-body HIIT (WBHIIT), offer significant benefits for mobility and spinal posture in sedentary women. Pilates, well established in the literature as an effective intervention for improving flexibility and correcting postural deviations, demonstrated the ability to increase range of motion (ROM) in the lumbar and thoracic regions during gait, particularly benefiting individuals with mobility restrictions. WBHIIT, on the other hand, showed promising results, emerging as a low-cost and easily applicable alternative to combat sedentary behaviour, with a positive impact on ROM and cardiorespiratory health. The adaptations promoted by these interventions in spinal posture, especially in the sagittal plane, are important for preventing postural issues and reducing the risks of spinal stiffness and pain, expanding the spectrum of preventive and therapeutic approaches to sedentary behaviour.

To further explore the effects of WBHIIT and Pilates on spinal mobility, future research should prioritise conducting randomised controlled trials (RCTs) that compare these interventions in terms of their impact on spinal mobility and posture. These studies should focus on diverse populations, including individuals with varying levels of physical fitness and different age groups, to examine how these modalities affect spinal health across different demographics. Additionally, the long-term effects of both interventions on spinal stiffness and pain should be investigated, along with their potential to reduce the risk of musculoskeletal disorders. Exploring the specific mechanisms by which WBHIIT and Pilates contribute to spinal mobility could provide valuable insights into their therapeutic potential.

## Figures and Tables

**Figure 1 ijerph-22-00162-f001:**
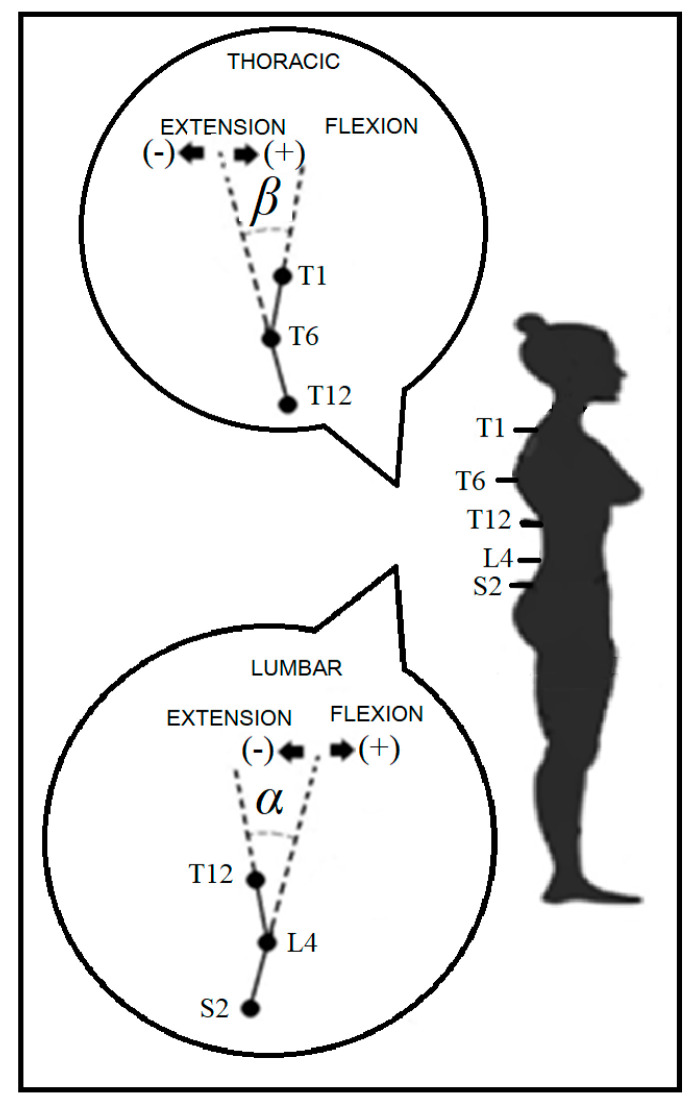
Lumbar and thoracic angles in the sagittal plane of the trunk. Legend: *α*: lumbar angle; *β*: thoracic angle; S2: second sacral vertebra; L4: fourth lumbar vertebra; T1, T6, T12: 1st, 6th, and 12th thoracic vertebrae.

**Figure 2 ijerph-22-00162-f002:**
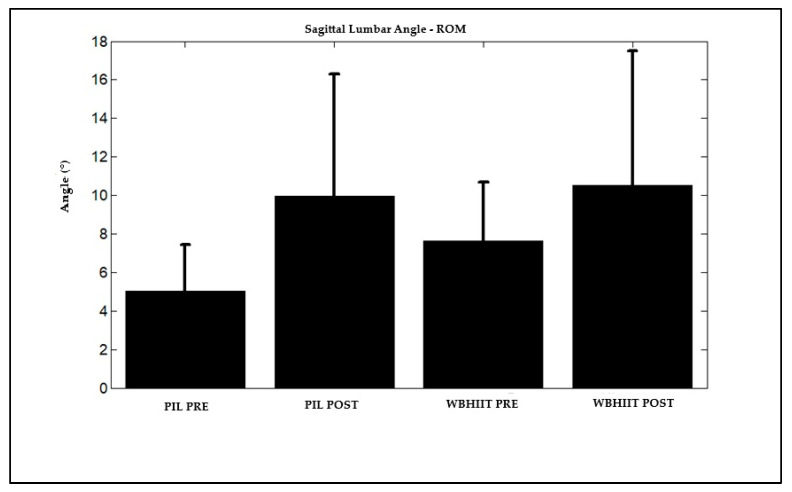
Range of motion of the spine in the lumbar region, in the sagittal plane, during walking (5 km/h) of Pilates and WBHIIT practitioners, before and after the intervention. Legend: ROM (range of motion during the gait cycle); WBHIIT: whole-body high-intensity interval training; PIL: Pilates; PRE: before intervention; POST: after intervention.

**Figure 3 ijerph-22-00162-f003:**
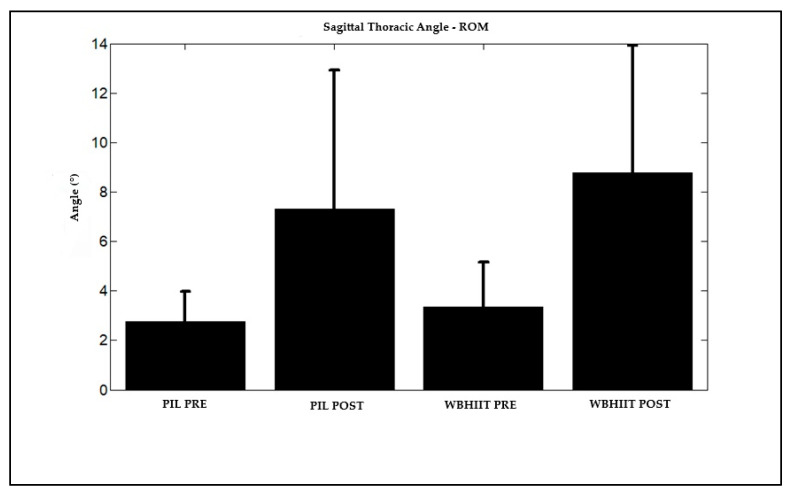
Range of motion of the spine in the thoracic region, in the sagittal plane, during walking (5 km/h) of Pilates and HWB practitioners, before and after the intervention. Legend: ROM (range of motion during the gait cycle); WBHIIT: whole-body high-intensity interval training; PIL: Pilates; PRE: before intervention; POST: after intervention.

**Table 1 ijerph-22-00162-t001:** Baseline characteristics of the volunteers.

Variables	Pilates (*n* = 13)	WBHIIT (*n* = 13)	Independent *t*-Test (*p*)
Age (years)	35.6 (12.1)	29.3 (8.9)	0.1421
BMI (kg/m^2^)	22.8 (4.3)	24.9 (3.4)	0.1844
WHR	0.75 (0.07)	0.77 (0.05)	0.5804
Waist Circumference (cm)	74.0 (9.0)	78.9 (5.8)	0.1161

Legend: Data presented as average (standard deviation). Abbreviations: *n*—group size; WBHIIT—whole-body high-intensity interval training; BMI—body mass index; WHR—waist-to-hip ratio.

**Table 2 ijerph-22-00162-t002:** Angles (°) of spine [average ± standard deviation] during gait (5 km/h) in Pilates and WBHIIT practitioners.

Variables	Pilates (*n* = 13)		WBHIIT (*n* = 13)		Anova 2 Way [*p/F*]			*Cohen’s d*	
	Pre	Post	Pre	Post	Group	Time	Interaction	PIL	WBHIIT
L-Static (°)	−27.3 ± 12.5	−30.8 ± 12.2	−38.7 ± 10.0	−28.6 ± 17.3	0.218/1.554	0.374/0.804	0.069/3.453		
L-ROM (°)	**5.0 ± 2.4**	**10.0 ± 6.3 ***	**7.6 ± 3.0**	**10.5 ± 7.0 ***	0.269/1.249	**0.008/7.651**	0.469/0.531	**1.8**	**1.0**
L-Aver (°)	−22.2 ± 12.5	−25.2 ± 9.5	−28.3 ± 11.2	−21.6 ± 12.6	0.696/0.155	0.572/0.325	0.135/2.305		
T-Static (°)	**14.9 ± 7.1 ^#^**	**15.7 ± 6.9 ^#^**	**21.8 ± 6.6**	**19.5 ± 7.0**	**0.007/7.862**	0.708/0.142	0.423/0.653		
T-ROM (°)	**2.8 ± 1.2**	**7.3 ± 5.6 ***	**3.3 ± 1.8**	**8.8 ± 5.1 ***	0.355/0.873	**0.000/20.71**	0.687/0.164	**2.9**	**3.5**
T-Aver (°)	15.4 ± 6.5	15.9 ± 6.2	20.1 ± 5.8	17.6 ± 6.9	0.080/3.209	0.585/0.303	0.402/0.715		

Legend: Bold values indicate statistically significant results. Static: standing position; ROM: range of motion in the gait cycle; Aver: average value presented in gait cycle; L: lumbar region; T: thoracic region; *: significantly different from pre-intervention; ^#^**^:^** significantly different from WBHIIT group; 2-way ANOVA: two-factor analysis of variance; *p* and *F*: ANOVA results; *Cohen’s d*: effect size; WBHIIT: whole-body high-intensity interval training; PIL: Pilates; *n*: group size; Pre: before intervention; Post: after intervention.

## Data Availability

The datasets presented in this article are not readily available for ethical reasons. Requests for access to datasets should be directed to mariohcampos@ufg.br.

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
