# Peer review of "Eight-Week Pilates or Whole-Body High-Intensity Interval Training Program Improves Spinal Range of Motion During the Gait Cycle in Sedentary Women: A Preliminary Study"

_ijerph, 2025, doi:10.3390/ijerph22020162_

Round 1
Reviewer 1 Report
Comments and Suggestions for Authors
The study compares two different exercise programs, Pilates and WBHIIT, aimed at reducing the negative effects of a sedentary lifestyle. Addressing sedentary behavior and improving spinal health are highly significant public health issues in today's world, making this intervention particularly relevant. However, there are some aspects I would like to critique as weaknesses of the study.
Firstly, the sample size is very small. This limited sample size restricts the statistical evaluation of the results and also hinders the ability to draw more definitive conclusions about the differences between the two methods. Additionally, a control group should have been included in the study.
While I do not think the study is poor, I believe it could benefit significantly from revisions. However, at this point, it seems unlikely that the sample size can be increased or a control group can be added.
Author Response
Indeed, the use of a control group and an increase in the sample size could enhance the methodological quality of the study. However, as the reviewer has noted, the study is now complete, and it is not possible to alter the sample. We have made an effort to highlight the limitations of this sample to the readers (in the discussion and conclusion) to prevent unwarranted generalizations. Despite these limitations, we believe that the study provides original data to the literature, making its publication valid with the noted caveats.
We also thank you for your valuable feedback on our work.
Reviewer 2 Report
Comments and Suggestions for Authors
Review comment
The study, entitled "Eight-Week Pilates or Whole-Body High-Intensity Interval Training Program Improves Spinal Range of Motion During the Gait Cycle in Sedentary Women", aims to compare the effects of Pilates and whole body high intensity interval training (WBHIIT) on spinal curvature in sedentary women. The study uses a non-randomized controlled trial of eight weeks of Pilates and whole-body high-intensity interval training for two separate groups of subjects. The results of the study indicates an increase in spinal range of motion in both groups, highlighting the importance of combining strengthening and mobility exercises to mitigate the effects of sedentary behavior and support spinal health. However, this study is limited by a small sample size, which affects its generalizability, and the interventions are not sufficiently detailed and require further additions. Therefore, a major revision is recommended. Detailed comments for each section follow.
Specific comments
1. In the Abstract part, there is no mention of the conclusions of this study. In the opinion of reviewer, the author needs to add this refinement at the end of the abstract in order to clarify the innovative aspects of the study. (Line19-20)
2. In my opinion, in the introduction part, there are fewer references to the background of the study, insufficient detail in the background information, insufficient references to previous work, and a lack of detail on the current level of development of the problem. I suggest that an addition should be made. (Line40)
3. “After being invited, voluntarily agreed to participate in the study 13 sedentary woman enrolled in Pilates and 13 in Whole Body HIIT (WBHIIT) classes offered by the Center for Body Practices at the University (Table 1).” Justification for such a small sample size is missing here. The authors could briefly explain the rationale, perhaps mentioning the exploratory nature of the study. Additionally, the relatively small sample size of the study (13 individuals per group) may affect the extrapolation of the results. It is recommended that the authors discuss the impact of the sample size on the findings of the study and consider conducting an efficacy analysis to determine whether there is a sufficient sample size to detect an actual effect size. (Line53-55)
4. In the Materials and methods part, “The Inclusion criteria were ages between 18 and 59, absence of regular exercise, and fitness for regular physical activity.” Age range varies significantly. How did the authors to control for potential biomechanical differences attributable to age? If this was not addressed, it may introduce variability in results. (Line64-65)
5. “Additionally, the method’s intensity and progression increased by raising the level of difficulty, resistance, and range of motion of each exercise when optimal execution and control were achieved.” In terms of interventions, it is recommended that the authors provide a more detailed description of the interventions, including the specific exercises, intensity, duration, and progression of Pilates and WBHIIT, so that other researchers can replicate these interventions.(Line95-97)
6. “Regarding gait (Table 2), there was no difference between groups for average angles as for range of motion. No differences were presented for walking average angles after interventions. However, both exercise modalities promoted a large magnitude increase in the range of motion of the lumbar and thoracic regions in the sagittal plane.” In the results part, it is recommended that the authors add graphics to more visually demonstrate key results, such as changes in spinal mobility.(Line196-199)
7. “Extrapolating these considerations to individuals with chronic low back pain, it can be inferred, based on the findings of Will et al. [40] and Seyedhoseinpoor et al. [41], that chronic pain in the lumbar region may result in the contraction of the paravertebral muscles and adjacent structures.” In my opinion, the statements like“it can be inferred”and “may result in” are too subjective. Consider rephrasing to make this statement more direct or supported by the data, as this will enhance clarity and scientific rigor.(Line246-248)
8. In the conclusion part, it is recommended that the authors suggest future research directions based on the current findings to further explore the effects of Pilates and WBHIIT on spinal mobility.
Author Response
We agree that it is necessary to add the conclusions to the abstract, and we have submitted the manuscript with the requested changes. Regarding previous studies, to the best of our knowledge, we have not found studies linking spinal range of motion and sedentary behavior in WBHIIT practitioners, only in Pilates practitioners. However, we conducted a thorough search and added more robust references to the review to enhance the discussion of the topic and problem presented. Indeed, the sample size is a limitation of the present study. Following the reviewer’s recommendation, we highlighted these sample limitations to readers to prevent unwarranted generalizations, as detailed in the manuscript sections: a) Materials and Methods - “This research is part of a larger project titled ‘Acute and Chronic Effects of Different Types of Physical Exercise on Biodynamic and Subjective Health Parameters.’ This project anticipates further phases with a larger sample size and random distribution of participants into groups for a more robust statistical design. The current work is the initial phase, using a convenience sample size. The interpretation of the results should consider these characteristics. Despite these limitations, the independent samples t-test (p < 0.05) showed baseline similarity between the groups (Table 1).”; b) Discussion - same text as in Materials and Methods; c) Conclusion - same text as in Materials and Methods. Despite these limitations, we believe the study provides original data to the literature, justifying its publication with the noted caveats. Additionally, we have corrected inaccuracies regarding the participants' characteristics. The ages ranged from 29 to 47 years, as shown in Table 1, and no significant differences in baseline age were observed. All participants were deemed fit for exercise through a socio-demographic questionnaire and were healthy, without neuromuscular, musculoskeletal, or joint diseases that would prevent participation in the exercise protocols. We have also added details of the interventions, as requested, highlighted in the text, and included figures demonstrating the post-intervention increase in range of motion. All indicated corrections have been made in the submitted material and are highlighted in the revised text.
We also thank you for your valuable feedback on our work.
Reviewer 3 Report
Comments and Suggestions for Authors
This study aimed to compare the effect of Pilates or Whole Body High-Intensity Interval Training (WBHIIT) variation, which utilizes bodyweight exercises, fostering cardiopulmonary health and posture improvements, on the spinal curvature of sedentary women (spending seven to eight hours seated daily). Twenty-six participants, aged 18 to 59, who fulfilled the inclusion criteria, were non-randomly divided into two groups (13 each) and underwent an eight-week intervention of either Pilates or WBHIIT. Women included in the Pilates group were older (34.50 [3.29] vs 29.31 [2.45], and had lower Body Mass Index (BMI): 22.80 (1.10) vs 24.91 (0.93). The Pilates intervention method was conducted over eight weeks, with 50-minute sessions twice weekly totaling 16 sessions. The training sessions were group-based according to the Authentic Pilates Studio Brazil. WBHIIT protocol was conducted over eight weeks and prescribed based on the study by Evangelista et al in 2021. During the first two weeks, 15 rounds of 30 seconds of 105 high-intensity exercise (>9 on the Subjective Perception of Effort - SPE - or “all out,” as many repetitions as possible) were performed, followed by 30 seconds of passive recovery. From the third week, the protocol was modified to 20 rounds of 30 seconds of high-intensity exercise (>9 on SPE or “all out”), followed by 30 seconds of active recovery (SPE <4) with a low-complexity, low-metabolic-demand exercise, such as walking or light jogging. The adapted Borg Scale for Subjective Perception of Effort was used in Brazil. The exercises used during the WBHIIT stimulus phase predominantly included push-ups, jumping jacks, high knee running in place, step climbing, squats, squat jumps, burpees, split squats, mountain climbers, alternating lunges, and jump alternating lunges. The training session lasted 40 minutes, including a 10-minute warm-up with low-impact aerobic exercises of moderate intensity, such as brisk walking or running in place, and a 10-minute cool-down with static stretching for the entire body. A total of WBHIIT sessions were completed over eight weeks.
Spinal posture was assessed pre- and post-intervention through videogrammetry during standing and walking positions. Results: The findings indicated that both groups displayed an increase in the spinal range of motion, highlighting the importance of incorporating strengthening and mobility exercises to mitigate sedentary behavior effects and support spinal health. There was no difference between groups for average angles as for range of motion during walking. No differences were presented for walking average angles after interventions. However, both exercise modalities promoted a large magnitude increase in the range of motion of the lumbar and thoracic regions in the sagittal plane.
Authors conclusion (included only in the main text, absent in the abstract): both modalities, Pilates and Whole Body HIIT (WBHIIT), offer significant benefits for mobility and spinal posture in sedentary women. Pilates, well-established in the literature as an effective intervention for improving flexibility and correcting postural deviations, demonstrated the ability to increase range of motion (ROM) in the lumbar and thoracic regions during gait, particularly benefiting individuals with mobility restrictions. WBHIIT, on the other hand, showed promising results, emerging as a low-cost and easily applicable alternative to combat sedentary behavior, with a positive impact on ROM and cardiorespiratory health. The adaptations promoted by these interventions in spinal posture, especially in the sagittal plane, are important for preventing postural issues and reducing the risks of spinal stiffness and pain, expanding the spectrum of preventive and therapeutic approaches to sedentary behavior.
Reviewer's opinion. This study seems to be very interesting, but the report needs some improvement. I suggest changing the title to: „The Effect of Pilates or Whole-Body High-Intensity Interval Training on Spinal Range of Motion in Sedentary Women – Preliminary Study” (or „pilot study”). The is necessary to prepare a longer abstract containing more information, especially on the lack of randomization. The lack of randomization lowers the value of work. The explanation given by the authors about the lack of randomization is not convincing. The abstract should be more informative and should contain also the author`s conclusions. The next question is: why was it not statistically calculated whether there is a relationship between changes in spine mobility and body mass index? Why was it not statistically calculated whether there is a relationship between changes in spine mobility and age?
Author Response
We have changed the title to "The Effect of an Eight-Week Pilates or Whole-Body High-Intensity Interval Training Program on Spinal Range of Motion During the Gait Cycle in Sedentary Women: A Preliminary Study." The sampling was by convenience, respecting the participants' preference to choose and practice the exercises they desired, which likely contributed to their adherence and retention in the chosen modality, thereby increasing the ecological validity of the study. However, we acknowledge the need for randomization, blinding, and a control group, and we have suggested in the revised text that future researchers in this area implement these measures for similar interventions. The objective of this study was not related to analyzing associations or correlations, but we appreciate this suggestion and will consider it for future articles. All indicated corrections have been made in the submitted material and are highlighted in the text.
We also thank you for your valuable feedback on our work.
Round 2
Reviewer 1 Report
Comments and Suggestions for Authors
I agree with the authors, sometimes it is not possible to make changes such as the number of samples after the study is completed. However, the writing part of the study has been improved.
Reviewer 2 Report
Comments and Suggestions for Authors
Thanks to the efforts of the authors during the revision process, the reviewer believes that the manuscript now meets the criteria for publication.
Reviewer 3 Report
Comments and Suggestions for Authors
In my first review, I suggested changing the title to: „… – Preliminary Study”... and it is done. In the revised manuscript the authors changed and added many phrases. They prepared a longer abstract containing more information. The lack of randomization lowers the value of work. A few positions of literature have been added to references.
Final decision: in this form, the revised manuscript deserves publication.